# AVERAGE REWARD REINFORCEMENT LEARNING WITH MONOTONIC POLICY IMPROVEMENT

## ABSTRACT

In continuing control tasks, an agent's average reward per time step is a more natural performance measure compared to the commonly used discounting framework since it can better capture an agent's long-term behavior. We derive a novel lower bound on the difference of the long-term average reward for two policies. The lower bound depends on the average divergence between the policies and on the so-called Kemeny constant, which measures to what degree the unichain Markov chains associated with the policies are well-connected. We also show that previous work based on the discounted return (Schulman et al., 2015; Achiam et al., 2017) results in a non-meaningful lower bound in the average reward setting. Based on our lower bound, we develop an iterative procedure which produces a sequence of monotonically improved policies for the average reward criterion. When combined with Deep Reinforcement Learning (DRL) methods, the procedure leads to scalable and efficient algorithms for maximizing the agent's average reward performance. Empirically we demonstrate the effectiveness of our method on continuing control tasks and show how discounting can lead to unsatisfactory performance.

## 1 INTRODUCTION

The goal of Reinforcement Learning (RL) is to build agents that can learn high-performing behaviors through trial-and-error interactions with the environment. Broadly speaking, modern RL tackles two kinds of problems: *episodic tasks* and *continuing tasks*. In episodic tasks, the agent-environment interaction can be broken into separate distinct episodes, and the performance of the agent is simply the sum of the rewards accrued within an episode. Examples of episodic tasks include training an agent to learn to play Go (Silver et al., 2016; 2018) or Atari video games (Mnih et al., 2013), where the episode terminates when the game ends. In continuing tasks, such as controlling robots with long operating lifespans (Peters & Schaal, 2008; Schulman et al., 2015; Haarnoja et al., 2018), there is no natural separation of episodes and the agent-environment interaction continues indefinitely. The performance of an agent in a continuing task is more difficult to quantify since even for bounded reward functions, the total sum of rewards is typically infinite.

One way of making the long-term reward objective meaningful for continuing tasks is to apply *discounting*, i.e., we maximize the discounted sum of rewards $r_0 + \gamma r_1 + \gamma^2 r_2 + \cdots$ for some discount factor $\gamma \in (0, 1)$. This is guaranteed to be finite for any bounded reward function. However the discounted objective biases the optimal policy to choose actions that lead to high near-term performance rather than to high long-term performance. Such an objective — while useful in certain applications — is not appropriate when the goal is optimize long-term behavior. As argued in Chapter 10 of Sutton & Barto (2018) and in Naik et al. (2019), a more natural objective is to use the average reward received by an agent over every time-step. While the average reward setting has been extensively studied in the classical Markov Decision Process literature (Howard, 1960; Blackwell, 1962; Veinott, 1966; Bertsekas et al., 1995), it is much less commonly used in reinforcement learning. An important open question is whether recent advances in RL for the discounted reward criterion can be naturally generalized to the average reward setting.

One major source of difficulty with modern DRL algorithms lies in controlling the step-size for policy updates. In order to have better control over step-sizes, Schulman et al. (2015) constructed a lower bound on the difference between the expected discounted return for two arbitrary policies $\pi$ and $\pi'$. The bound is a function of the divergence between these two policies and the discount factor.

Schulman et al. (2015) showed that iteratively maximizing this lower bound generates a sequence of monotonically improved policies in terms of their discounted return.

In this paper, we first show that the policy improvement theorem from Schulman et al. (2015) results in a non-meaningful bound in the average reward case. We then derive a novel result which lower bounds the difference of the average rewards based on the divergence of the policies. The bound depends on the average divergence between the policies and on the so-called Kemeny constant, which measures to what degree the unichain Markov chains associated with the policies are well-connected. We show that iteratively maximizing this lower bound guarantees monotonic average reward policy improvement. Similar to the discounted case, the problem of maximizing the lower bound can be approximated with DRL algorithms which can be optimized using samples collected in the environment. We describe in detail two such algorithms: Average Reward TRPO (ATRPO) and Average Cost CPO (ACPO), which are average reward versions of algorithms based on the discounted criterion (Schulman et al., 2015; Achiam et al., 2017). Using the MuJoCo simulated robotic benchmark, we carry out extensive experiments with the ATRPO algorithm and show that it is more effective than their discounted counterparts for these continuing control tasks. To our knowledge, this is one of the first paper to address DRL using the long-term average reward criterion.

## 2 PRELIMINARIES

Consider a Markov Decision Process (MDP) (Sutton & Barto, 2018) $(\mathcal{S}, \mathcal{A}, P, r, \mu)$ where the state space $\mathcal{S}$ and action space $\mathcal{A}$ are assumed to be finite. The transition probability is denoted by $P : \mathcal{S} \times \mathcal{A} \times \mathcal{S} \to [0, 1]$, the bounded reward function $r : \mathcal{S} \times \mathcal{A} \to [r_{\min}, r_{\max}]$, and $\mu : \mathcal{S} \to [0, 1]$ is the initial state distribution. Let $\pi = \{\pi(a|s) : s \in \mathcal{S}, a \in \mathcal{A}\}$ be a stationary policy, and $\Pi$ is the set of all stationary policies. Here we discuss the two objective formulations for continuing control tasks: the average reward approach and discounted reward approach.

**Average Reward Approach**
In this paper, we will focus exclusively on *unichain* MDPs, which is when the Markov chain corresponding to every policy contains only one recurrent class and a finite but possibly empty set of transient states. The average reward objective is defined as:

$$\rho(\pi) := \lim_{N \to \infty} \frac{1}{N} \mathop{\mathbb{E}}_{\tau \sim \pi} \left[ \sum_{t=0}^{N-1} r(s_t, a_t) \right] = \mathop{\mathbb{E}}_{\substack{s \sim d_\pi \\ a \sim \pi}} [r(s, a)]. \tag{1}$$

Here $d_\pi(s) := \lim_{N \to \infty} \frac{1}{N} \sum_{t=0}^{N-1} P(s_t = s | \pi) = \lim_{t \to \infty} P(s_t = s | \pi)$ is the *stationary state distribution under policy* $\pi$, $\tau = (s_0, a_0, \dots, )$ is a sample trajectory. We use $\tau \sim \pi$ to indicate that the trajectory is sampled from policy $\pi$, i.e. $s_0 \sim \mu$, $a_t \sim \pi(\cdot|s_t)$, and $s_{t+1} \sim P(\cdot|s_t, a_t)$. In the unichain case, the average reward $\rho(\pi)$ is state-independent for any policy $\pi$ (Bertsekas et al., 1995).

We express the *average-reward value function* as $V^\pi(s) := \mathbb{E}_{\tau \sim \pi} \left[ \sum_{t=0}^\infty (r(s_t, a_t) - \rho(\pi)) \Big| s_0 = s \right]$ and *action-value function* as $Q^\pi(s, a) := \mathbb{E}_{\tau \sim \pi} \left[ \sum_{t=0}^\infty (r(s_t, a_t) - \rho(\pi)) \Big| s_0 = s, a_0 = a \right]$. We define the *average reward advantage function* as $A^\pi(s, a) := Q^\pi(s, a) - V^\pi(s)$.

**Discounted Reward Approach**
For some discount factor $\gamma \in (0, 1)$, the discounted reward objective is defined as

$$\rho_\gamma(\pi) := \mathop{\mathbb{E}}_{\tau \sim \pi} \left[ \sum_{t=0}^\infty \gamma^t r(s_t, a_t) \right] = \frac{1}{1 - \gamma} \mathop{\mathbb{E}}_{\substack{s \sim d_{\pi,\gamma} \\ a \sim \pi}} [r(s, a)]. \tag{2}$$

where $d_{\pi,\gamma}(s) := (1 - \gamma) \sum_{t=0}^\infty \gamma^t P(s_t = s | \pi)$ is known as the *future discounted state visitation distribution under policy* $\pi$. Note that unlike the average reward objective, *the discounted objective depends on the initial state distribution* $\mu$. It can be easily shown that $d_{\pi,\gamma}(s) \to d_\pi(s)$ for all $s$ as $\gamma \to 1$. The *discounted value function* is defined as $V_\gamma^\pi(s) := \mathbb{E}_{\tau \sim \pi} \left[ \sum_{t=0}^\infty \gamma^t r(s_t, a_t) \Big| s_0 = s \right]$ and *discounted action-value function* $Q_\gamma^\pi(s, a) := \mathbb{E}_{\tau \sim \pi} \left[ \sum_{t=0}^\infty \gamma^t r(s_t, a_t) \Big| s_0 = s, a_0 = a \right]$. Finally, the *discounted advantage function* is defined as $A_\gamma^\pi(s, a) := Q_\gamma^\pi(s, a) - V_\gamma^\pi(s)$.

It is well-known that $\lim_{\gamma \to 1}(1-\gamma)\rho_\gamma(\pi) = \rho(\pi)$, implying that the discounted and average reward objectives are equivalent in the limit as $\gamma$ approaches 1 (Blackwell, 1962). We will further discuss the relationship between the discounted and average reward value functions in the supplementary materials and prove that $\lim_{\gamma \to 1} A_\gamma^\pi(s,a) = A^\pi(s,a)$ (see Corollary A.1).

## 3 MONTONICALLY IMPROVEMENT GUARANTEES FOR DISCOUNTED RL

In many modern RL literature (Schulman et al., 2015; 2017; Abdolmaleki et al., 2018; Vuong et al., 2019), algorithms iteratively update policies within a local region, i.e., at iteration $k$ we find policy $\pi_{k+1}$ by maximizing $\rho_\gamma(\pi)$ within some region $D(\pi, \pi_k) \leq \delta$ for some divergence measure $D$. This approach allows us to control the step-size of each update using different choices of $D$ and $\delta$ which can lead to better sample efficiency (Peters & Schaal, 2008). Schulman et al. (2015) derived a policy improvement bound based on a specific choice of $D$:

$$\rho_\gamma(\pi_{k+1}) - \rho_\gamma(\pi_k) \geq \frac{1}{1-\gamma} \mathop{\mathbb{E}}_{\substack{s \sim d_{\pi_k,\gamma} \\ a \sim \pi_{k+1}}} [A_\gamma^{\pi_k}(s,a)] - C \cdot \max_s [D_{\mathrm{TV}}(\pi_{k+1} \parallel \pi_k)[s]] \tag{3}$$

where $D_{\mathrm{TV}}(\pi' \parallel \pi)[s] := \frac{1}{2}\sum_a |\pi'(a|s) - \pi(a|s)|$ is the *total variation divergence* for policies $\pi$ and $\pi'$, and $C$ is some constant which does not depend on the divergence term $D_{\mathrm{TV}}$. Schulman et al. (2015) showed that by choosing $\pi_{k+1}$ such that the right hand side of (3) is maximized, we are guaranteed to have $\rho_\gamma(\pi_{k+1}) \geq \rho_\gamma(\pi_k)$. This provided the theoretical foundation for an entire class of scalable policy optimization algorithms based on efficiently maximizing the right-hand-side of (3) (Schulman et al., 2015; 2017; Wu et al., 2017; Abdolmaleki et al., 2018; Vuong et al., 2019).

A natural question arises here is whether the iterative procedure described by Schulman et al. (2015) also guarantees improvement w.r.t. the average reward. Since the discounted and average reward objectives are equivalent when $\gamma \to 1$, one may assume that we can also lower bound the policy performance difference of the average reward objective by letting $\gamma \to 1$ for the bounds in Schulman et al. (2015). Unfortunately this results in a non-meaningful bound. We will demonstrate this through a similar policy improvement bound from Achiam et al. (2017) based on the average divergence but a similar argument can be made for the original bound from Schulman et al. (2015) (see supplementary material for proof and discussion).

**Proposition 1.** *Consider the following bound from Achiam et al. (2017)*

$$D_{\pi,\gamma}^-(\pi') \leq \rho_\gamma(\pi') - \rho_\gamma(\pi) \leq D_{\pi,\gamma}^+(\pi') \tag{4}$$

*where*

$$D_{\pi,\gamma}^\pm(\pi') = \frac{1}{1-\gamma} \mathop{\mathbb{E}}_{\substack{s \sim d_\pi \\ a \sim \pi}} \left[ \frac{\pi'(a|s)}{\pi(a|s)} A_\gamma^\pi(s,a) \right] \pm \frac{2\gamma\epsilon_\gamma}{(1-\gamma)^2} \mathop{\mathbb{E}}_{s \sim d_\pi} [D_{TV}(\pi' \parallel \pi)[s]]$$

*and $\epsilon_\gamma = \max_s \left| \mathbb{E}_{a \sim \pi'}[A_\gamma^\pi(s,a)] \right|$. We have:*

$$\lim_{\gamma \to 1}(1-\gamma)D_{\pi,\gamma}^\pm(\pi') = \pm\infty \tag{5}$$

Since $\lim_{\gamma \to 1}(1-\gamma)(\rho_\gamma(\pi') - \rho_\gamma(\pi)) = \rho(\pi') - \rho(\pi)$, Proposition 1 says (4) becomes trivial when used on the average reward. This result is discouraging as it shows that the policy improvement guarantee from Schulman et al. (2015) does not appear to generalize to the average reward setting. In the next section, we will derive an alternative policy improvement bound for the average reward objective which can be used to generate monotonically improved policies w.r.t. the average reward.

## 4 MAIN RESULTS

### 4.1 AVERAGE REWARD POLICY IMPROVEMENT THEOREM

Let $d_\pi \in \mathbb{R}^{|\mathcal{S}|}$ be the probability column vector whose components are $d_\pi(s)$, $P_\pi \in \mathbb{R}^{|\mathcal{S}| \times |\mathcal{S}|}$ be the transition matrix under policy $\pi$ whose $(s, s')$ component is $P_\pi(s'|s) = \sum_a P(s'|s,a)\pi(a|s)$, and $P_\pi^* = \lim_{t \to \infty} P_\pi^t$ be the limiting distribution for the transition matrix. We use $\|\cdot\|_p$ to denote the

operator norm of a matrix. In particular $\|\cdot\|_1$ and $\|\cdot\|_\infty$ are the maximum absolute column sum and maximum absolute row sum of a matrix respectively (Horn & Johnson, 2012).

Suppose we have a new policy $\pi'$ obtained via some update rule from the current policy $\pi$. Similar to the discounted case, we would like to measure their performance difference $\rho(\pi') - \rho(\pi)$ using an expression which depends on $\pi$ and some divergence metric between the two policies. The following identity shows that $\rho(\pi') - \rho(\pi)$ can be expressed using the advantange function of $\pi$.

**Lemma 1.** *For any two stochastic policies $\pi$ and $\pi'$:*

$$\rho(\pi') - \rho(\pi) = \lim_{N \to \infty} \frac{1}{N} \mathop{\mathbb{E}}_{\tau \sim \pi'} \left[ \sum_{t=0}^{N-1} A^\pi(s_t, a_t) \right] = \mathop{\mathbb{E}}_{\substack{s \sim d_{\pi'} \\ a \sim \pi'}} [A^\pi(s, a)] \tag{6}$$

Lemma 1 is the an extension of the well-known policy difference lemma from Kakade & Langford (2002) to the average reward case. A similar result was proved by Neu et al. (2010) and Even-Dar et al. (2009). For completeness, We will provide a proof based on the Bellman equation as well as a simpler alternative proof in the supplementary material. Note that this expression depends on samples drawn from $\pi'$. However we can show through the following lemma that when $d_\pi$ and $d_{\pi'}$ are "close," we can evaluate the expression in (6) using samples from $d_\pi$ (see supplementary material for proof).

**Lemma 2.** *For any two stochastic policies $\pi$ and $\pi'$, the following bound holds:*

$$\mathop{\mathbb{E}}_{\substack{s \sim d_\pi \\ a \sim \pi'}} [A^\pi(s, a)] - 2\epsilon D_{TV}(d_{\pi'} \| d_\pi) \leq \rho(\pi') - \rho(\pi) \leq \mathop{\mathbb{E}}_{\substack{s \sim d_\pi \\ a \sim \pi'}} [A^\pi(s, a)] + 2\epsilon D_{TV}(d_{\pi'} \| d_\pi) \tag{7}$$

*where $\epsilon = \max_s \left| \mathbb{E}_{a \sim \pi'(a|s)}[A^\pi(s, a)] \right|$.*

Lemma 2 shows us how policy improvement is related to the stationary distribution underlying each policy. In order to study how policy improvement is connected to changes in the actual policies themselves, we need to analyze the relationship between changes in the policies and changes in stationary distributions. It turns out that the sensitivity of the stationary distributions in relation to the policies is related to the structure of the underlying Markov chain.

Let $M^\pi \in \mathbb{R}^{|\mathcal{S}| \times |\mathcal{S}|}$ be the *mean first passage time matrix* whose elements $M^\pi(s, s')$ is the expected number of steps it takes to reach state $s'$ from $s$ under policy $\pi$. The matrix $M^\pi(s, s')$ can be calculated via (Theorem 4.4.7 of Kemeny & Snell (1960))

$$M^\pi(s, s') = (I - Z^\pi + E Z^\pi_{\text{dg}}) D^\pi \tag{8}$$

where $Z^\pi = (I - P_\pi + P_\pi^*)^{-1}$ is known as the *fundamental matrix of the Markov chain* (Kemeny & Snell, 1960), $E$ is a square matrix consisting of all ones. The subscript 'dg' for some square matrix $A$ refers to a diagonal matrix whose elements are the diagonals of $A$. $D^\pi \in \mathbb{R}^{|\mathcal{S}| \times |\mathcal{S}|}$ is a diagonal matrix whose elements are $1/d_\pi(s)$. One important property of mean first passage time is that given some policy $\pi$:

$$\kappa^\pi = \sum_{s'} d_\pi(s') M^\pi(s, s') \tag{9}$$

is a constant independent of the starting state $s$. This result is known as the *random target lemma* (Aldous & Fill, 1995). The constant $\kappa^\pi$ is sometimes referred to as *Kemeny's constant* (Grinstead & Snell, 2012). This constant can be interpreted as the mean number of steps it takes to get to any goal state weighted by the steady-distribution of the goal states. This weighted mean does not depend on the starting state, as mentioned just above. The constant uses a single number to summarize how "well-connected" a Markov chain is. It can also be shown that $\kappa^\pi = \text{trace}(Z^\pi)$ (Grinstead & Snell, 2012). We then have the following result which connects the sensitivity of the stationary distribution to changes to the policy.

**Lemma 3.** *The divergence between the stationary distributions $d_\pi$ and $d_{\pi'}$ can be upper bounded by the average divergence between policies $\pi$ and $\pi'$ as follows:*

$$D_{TV}(d_{\pi'} \| d_\pi) \leq (\kappa^{\pi'} - 1) \mathop{\mathbb{E}}_{s \sim d_\pi} [D_{TV}(\pi' \| \pi)[s]] \tag{10}$$

We wish to point out here that Achiam et al. (2017) showed a similar result to Lemma 3 in the discounted case where the change in $d_{\pi,\gamma}$ can be bounded in terms of the change in policy up to a multiplicative constant which only depends on the discount factor. In the discounted case, this is possible since a discounted MDP is like a finite-horizon MDP problem; in fact, it can be shown to be equivalent to a related finite horizon problem (Proposition 5.3.1, Puterman (1994)). The discount factor can be used to control the effective horizon where larger discount factors correspond to longer horizons. In fact, it can be easily shown that the multiplicative factor from Achiam et al. (2017) goes to infinity as $\gamma \to 1$, meaning that the bound is not useful for long horizon problems. In the average reward setting, the sensitivity of the stationary distribution with respect to the policy can vary depending on the chain structure and long-term behavior of the underlying Markov chain. This means that it is only natural that the multiplicative constant in Lemma 3 depends on the transition matrix.

This result is also highly intuitive, For very "well-connected" Markov chains where an agent can easily and quickly get to any state, this constant is relatively small and the stationary distributions are not sensitive to small changes in policy. On the other hand, for Markov chains that are "weakly connected," where on average, it can take a long time to get to some recurrent state in the state space, the factor can become very large. In this case small changes in the policy can have a large impact on the resulting stationary distributions.

The following theorem connects the average reward performance of two policies and their average divergence.

**Theorem 1.** *For any two stochastic policies $\pi$ and $\pi'$, the following bounds hold:*

$$\rho(\pi') - \rho(\pi) \leq \underset{\substack{s \sim d_\pi \\ a \sim \pi}}{\mathbb{E}} \left[ \frac{\pi'(a|s)}{\pi(a|s)} A^\pi(s,a) \right] + 2\xi \underset{s \sim d_\pi}{\mathbb{E}} [D_{TV}(\pi' \parallel \pi)[s]] \tag{11}$$

$$\rho(\pi') - \rho(\pi) \geq \underset{\substack{s \sim d_\pi \\ a \sim \pi}}{\mathbb{E}} \left[ \frac{\pi'(a|s)}{\pi(a|s)} A^\pi(s,a) \right] - 2\xi \underset{s \sim d_\pi}{\mathbb{E}} [D_{TV}(\pi' \parallel \pi)[s]] \tag{12}$$

*where $\xi = (\kappa^{\pi'} - 1) \max_s \mathbb{E}_{a \sim \pi'} |A^\pi(s,a)|$.*

*Proof.* Combine the bounds from Lemma 2 and Lemma 3. Then rewrite the expectation for $A^\pi(s,a)$ as an expectation w.r.t. $\pi$ using importance sampling gives us the desired bound. $\square$

The right-hand-side of the bounds in Theorem 1 are guaranteed to be finite. Similar to the discounted case, the multiplicative factor $\xi$ provides a theoretical guidance on the step-sizes for policy updates (Schulman et al., 2015). The bound in Theorem 1 is given in terms of the TV divergence, however the KL divergence is more commonly used in practice. Vuong et al. (2019) compared various divergence measures and showed that the KL has superior empirical performance. The relationship between the TV divergence and KL divergence is given by Pinsker's inequality (Tsybakov, 2008), which says that for any two distributions $p$ and $q$: $D_{TV}(p \parallel q) \leq \sqrt{D_{KL}(p\|q)/2}$. We can then show that

$$\underset{s \sim d_\pi}{\mathbb{E}} [D_{TV}(\pi' \parallel \pi)[s]] \leq \underset{s \sim d_\pi}{\mathbb{E}} [\sqrt{D_{KL}(\pi'\|\pi)[s]/2}] \leq \sqrt{\underset{s \sim d_\pi}{\mathbb{E}}[D_{KL}(\pi'\|\pi)][s]]/2} \tag{13}$$

where the second inequality comes from Jensen's inequality. The inequality in (13) shows that the bounds in Theorem 1 still hold when $\mathbb{E}_{s \sim d_\pi}[D_{TV}(\pi' \parallel \pi)[s]]$ is substituted with $\sqrt{\mathbb{E}_{s \sim d_\pi}[D_{KL}(\pi'\|\pi)][s]/2}$.

### 4.2 Approximate Policy Iteration

One direct consequence of Theorem 1 is that iteratively maximizing the right-hand-side of (12) generates a monotonically improving sequence of policies w.r.t. the average reward objective. Algorithm 1 gives an approximate policy iteration algorithm that produces such a sequence of policies.

**Proposition 2.** *Given an initial policy $\pi_0$, Algorithm 1 is guaranteed to generate a sequence of policies $\pi_1, \pi_2, \ldots$ such that $\rho(\pi_0) \leq \rho(\pi_1) \leq \rho(\pi_2) \leq \cdots$.*

*Proof.* At iteration $k$, $\mathbb{E}_{s \sim d_{\pi_k}, a \sim \pi}[A^{\pi_k}(s,a)] = 0$, $\mathbb{E}_{s \sim d_{\pi_k}}[D_{KL}(\pi\|\pi_k)[s]] = 0$ for $\pi = \pi_k$. By Equation (14) and Theorem 1, $\rho(\pi_{k+1}) - \rho(\pi_k) \geq 0$. $\square$

---

**Algorithm 1** Approximate Policy Iteration for Average Reward Objective

---
**Initialize:** $\pi_0$
 1: **for** $k = 0, 1, 2, \ldots$ **do**
 2:     Policy evaluation step: evaluate $A^{\pi_k}(s, a)$ for all $s, a$.
 3:     Policy improvement step:

$$\pi_{k+1} = \underset{\pi}{\operatorname{argmax}} \left( \underset{\substack{s \sim d_{\pi_k} \\ a \sim \pi}}{\mathbb{E}} [A^{\pi_k}(s, a)] - \xi \sqrt{2 \underset{s \sim d_{\pi_k}}{\mathbb{E}} [D_{\mathrm{KL}}(\pi \| \pi_k)[s]]} \right) \tag{14}$$

where $\xi = (\kappa^\pi - 1) \max_s \mathbb{E}_{a \sim \pi} |A^{\pi_k}(s, a)|$

---

However, Algorithm 1 is difficult to implement in practice since it requires exact knowledge of the advantage function and transition matrix. Furthermore, calculating the term $\xi$ is impractical for high dimensional problems. In the next section, we will introduce a sample-based algorithm which approximates the update rule given in Equation (14).

## 5    PRACTICAL APPLICATIONS

As we have noted in the previous section, Algorithm 1 is not practical for problems with large state and action spaces and thus cannot be naïvely applied directly. In this section, we will discuss how Algorithm 1 and Theorem 1 can be used in practice to create algorithms which can effectively solve high dimensional DRL problems. In the Appendix C, we will also discuss how Theorem 1 can be used to solve DRL problems with safety constraints.

### 5.1    AVERAGE REWARD TRUST REGION POLICY OPTIMIZATION

For DRL problems, it is common to consider some parameterized policy class $\Pi_\Theta \subseteq \Pi$. Our goal is to devise a computationally tractable version of Algorithm 1 for policies in $\Pi_\Theta$, i.e., given a policy $\pi_{\theta_k}$ at iteration $k$, how do we obtain the best possible $\pi_{\theta_{k+1}}$? We can rewrite the unconstrained optimization problem in (14) as a constrained problem:

$$\underset{\pi_\theta \in \Pi_\theta}{\operatorname{maximize}} \underset{\substack{s \sim d_{\pi_{\theta_k}} \\ a \sim \pi_\theta}}{\mathbb{E}} [A^{\pi_{\theta_k}}(s, a)] \quad \text{s.t.} \quad \bar{D}_{\mathrm{KL}}(\pi_\theta \| \pi_{\theta_k}) \leq \delta \tag{15}$$

where $\bar{D}_{\mathrm{KL}}(\pi_\theta \| \pi_{\theta_k}) := \mathbb{E}_{s \sim d_{\pi_{\theta_k}}}[D_{\mathrm{KL}}(\pi_\theta \| \pi_{\theta_k})[s]]$. The constraint set $\{\pi_\theta \in \Pi_\Theta : \bar{D}_{\mathrm{KL}}(\pi_\theta \| \pi_{\theta_k}) \leq \delta\}$ is called the *trust region set*. This problem can be regarded as an average reward variant of TRPO from Schulman et al. (2015). Note that the advantage function in (15) is the *average reward advantage function* introduced in Section 2. When we set $\pi_{\theta_{k+1}}$ to be the optimal solution to (15), $\pi_{\theta_{k+1}}$ can be shown to have the following performance guarantee:

**Proposition 3.** *Let $\pi_{\theta_{k+1}}$ be the optimal solution to* (15) *for some $\pi_{\theta_k} \in \Pi_\Theta$. The policy performance difference between $\pi_{\theta_{k+1}}$ and $\pi_{\theta_k}$ can be lower bounded by*

$$\rho(\pi_{\theta_{k+1}}) - \rho(\pi_{\theta_k}) \geq -\xi^{\pi_{\theta_{k+1}}} \sqrt{2\delta} \tag{16}$$

*where $\xi^{\pi_{\theta_{k+1}}} = (\kappa^{\pi_{\theta_{k+1}}} - 1) \max_s \mathbb{E}_{a \sim \pi_{\theta_{k+1}}} |A^{\pi_{\theta_k}}(s, a)|$.*

*Proof.* Since $\bar{D}_{\mathrm{KL}}(\pi_{\theta_k} \| \pi_{\theta_k}) = 0$, $\pi_{\theta_k}$ is a feasible solution. The objective value is 0 for $\pi_\theta = \pi_{\theta_k}$. The bound follows from (12) and (13) where the average KL is bounded by $\delta$. □

Several algorithms have been proposed for efficiently solving the discounted version of (15): Schulman et al. (2015) and Wu et al. (2017) converts (15) into a convex problem via Taylor approximations; another approach is to first solve (15) in the nonparametric policy space and then project the result back into the parameter space (Abdolmaleki et al., 2018; Vuong et al., 2019). These algorithms can be adapted for the average reward case and are theoretically justified via Theorem 1 and Proposition 3. One notable difference compared to the discounted case is the estimation of the critic, as discussed in the next section and in the Appendix D.

## 5.2 IMPLEMENTATION

In this section, we discuss how the average reward version of the TRPO algorithm (Schulman et al., 2015) — which we will refer to as ATRPO — can be implemented in practice. Algorithm 2 provides a basic outline of the ATRPO algorithm.

---

**Algorithm 2** Average Reward TRPO (ATRPO)

---

**Initialize:** Policy parameters $\theta_0$, value net parameters $\phi_0$, learning rate $\alpha$.

1: **for** $k = 0, 1, 2, \cdots$ **do**
2:     Collect a sample trajectory $\{s_t, a_t, s_{t+1}, r_t\}$, $t = 1, \ldots, N$ from the environment using $\pi_{\theta_k}$.
3:     Calculate sample average reward of $\pi_{\theta_k}$ via $\rho = \frac{1}{N} \sum_{t=1}^{N} r_t$.
4:     **for** $t = 1, 2, \ldots, N$ **do**
5:         Get target $V_t^{\text{target}} = r_t - \rho + V_{\phi_k}(s_{t+1})$
6:         Get advantage estimate $\hat{A}(s_t, a_t) = r_t - \rho + V_{\phi_k}(s_{t+1}) - V_{\phi_k}(s_t)$
7:     Update critic by
$$\phi_{k+1} \leftarrow \phi_k - \alpha \nabla_\phi \mathcal{L}(\phi_k)$$
    where
$$\mathcal{L}(\phi_k) = \frac{1}{2} \sum_{t=1}^{N} \left\| V_{\phi_k}(s_t) - V_t^{\text{target}} \right\|^2$$
8:     Use $\hat{A}(s_t, a_t)$ to update $\theta_k$ using TRPO policy update (Schulman et al., 2015).

---

The major difference between the TRPO algorithm and the ATRPO algorithm is how the target for the critic and the advantage function are calculated. Importantly, simply letting $\gamma \to 1$ in TRPO does not lead to Algorithm 2. This subtle but important difference leads to a significant improvement in sample efficiency, as shown in the section on experimental results.

In Algorithm 2, for illustrative purposes, we use the average reward one-step bootstrapped estimate for the target of the critic and the advantage function. In practice, we instead use an average reward version of the Generalized Advantage Estimator (GAE) from Schulman et al. (2016). In short, GAE uses a tunable eligibility trace parameter $\lambda$ to act as a trade-off between the Monte Carlo estimate and the bootstrapped estimate. In the Appendix D we provide more detail on how GAE can be generalized to the average reward case.

## 6 RELATED WORK

Dynamic programming algorithms for finding the optimal average reward policies have been well-studied (Howard, 1960; Blackwell, 1962; Veinott, 1966). In contrary to our method which is based on the policy gradient approach, several Q-learning-like algorithms for problems with unknown dynamics have been proposed, such as R-Learning (Schwartz, 1993), RVI Q-Learning (Abounadi et al., 2001), and CSV-Learning (Yang et al., 2016). Mahadevan (1996) conducted a thorough empirical analysis of the R-Learning algorithm. We note that much of the previous work on average reward RL focuses on the tabular setting without function approximations, and the theoretical properties of many of these Q-learning-based algorithm are not well understood (in particular R-learning). More recently, POLITEX updates policies using a Boltzmann distribution over the sum of action-value function estimates of the previous policies (Abbasi-Yadkori et al., 2019) and Wei et al. (2020) introduced a model-free algorithm for optimizing the average reward of weakly-communicating MDPs. Both methods are shown to have theoretical guarantees under the tabular setting.

For policy gradient methods, Baxter & Bartlett (2001) showed that if $1/(1 - \gamma)$ is large compared to the mixing time of the Markov Chain induced by the MDP, then the gradient of $\rho_\gamma(\pi)$ can accurately approximate the gradient of $\rho(\pi)$. Kakade (2001) extended upon this result and provided an error bound on using an optimal discounted policy to maximize the average reward. In contrast, our work directly deals with using policy gradient methods for the average reward objective and provides theoretical guidance on the optimal step size for each policy update.

Policy improvement bounds have been extensively explored in the discounted case. The results from Schulman et al. (2015) is an extension of Kakade & Langford (2002) which restricted the policy class

to a mixture of policies. Pirotta et al. (2013) also proposed an alternative generalization to Kakade & Langford (2002). Achiam et al. (2017) improved upon Schulman et al. (2015) by replacing the maximum divergence with the average divergence.

# 7 EXPERIMENTS

Recently, DRL algorithms such as TRPO have proven to be successful for episodic high-dimensional tasks. In our experiments, we wish to study whether for continuing-control tasks, the policy trained with ATRPO can out-perform the policies trained with TRPO with different discount factors.

Our design goal for the experiments is to simulate continuing-control tasks where the agent can interact with the environment indefinitely. We consider three tasks (Ant, HalfCheetah, and Humanoid) from the MuJoCo physical simulator (Todorov et al., 2012) implemented in the OpenAI gym (Brockman et al., 2016). The natural goal is to train the agents to run as fast as possible without falling. However the standard MuJoCo tasks are episodic tasks which terminate when the agent falls. We convert these tasks into continuing control tasks via the following: when the agent falls, the agent incurs a large cost for falling, but then continues the trajectory from a random start state. We use these continuing-control tasks for both training and evaluation for both ATRPO and TRPO. More details on the environment can be found in Appendix F.

One point we wish to emphasize regarding the experiments is that even though the MuJoCo benchmark is commonly trained using the discounted objective (see e.g. Schulman et al. (2015), Wu et al. (2017), Schulman et al. (2017), Abdolmaleki et al. (2018), Vuong et al. (2019)), it is *always* evaluated using the undiscounted objective. This is because the undiscounted objective more naturally describes the goals of the MuJoCo agents (e.g., an agent's performance w.r.t. the reward signal should be equally important at time step 1000 as it is at time step 1). In the case of TRPO (and similarly many other DRL algorithms), discounting is used during training often for mathematical and computational convenience. Prior to our work, there has been no theoretical or empirical evidence to support applying trust region methods to the average reward. In this section, we demonstrate that when the actual objective we want to evaluate is undiscounted, discounting, as is commonly done, is unnecessary and may lead to suboptimal performance.

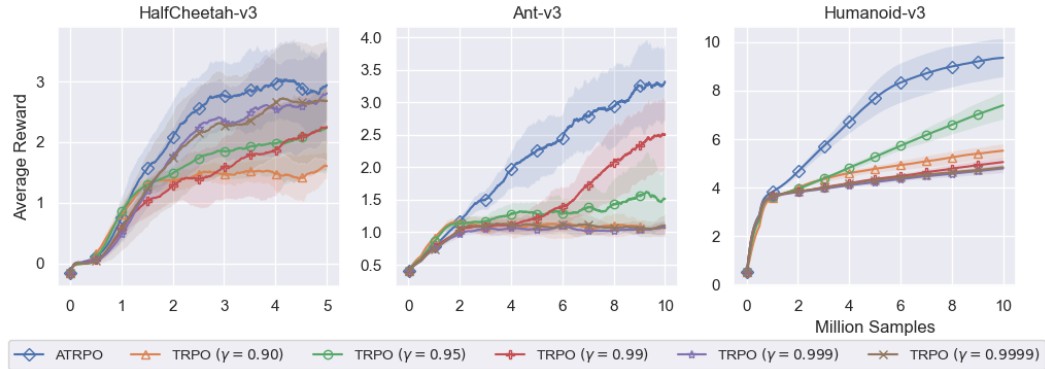

Figure 1: Learning curves comparing ATRPO and TRPO with different discount factors. The solid lines represent the average reward of trajectories of fixed length of 10,000 time steps averaged over the last 50 trajectories. The results are averaged over 10 random seeds and the shaded region represents one standard deviation.

During training, we collect one trajectory of a fixed length of 10,000 using the current policy.[1] We then use this data to update the critic and policy networks (see Algorithm 2). This gives us a new policy and critic which we then use to repeat the above process. In Figure 1, we plot the

---

[1]In the original OpenAI gym version of MuJoCo, episode lengths are capped at 1000 (see `https://github.com/openai/gym/blob/master/gym/envs/__init__.py`). We removed this cap to allow for arbitrarily long time horizons.

training curves of ATRPO and of TRPO for different discount factors. Detailed specifications and hyperparameter settings can be found in Appendix F.

Figure 1 shows that ATRPO improves performance by 5.0%, 32.8%, 26.7% on HalfCheetah, Ant and Humanoid respectively over TRPO with its best discount factors. One point worth noting is that increasing the discount factor does not necessarily lead to better performance of TRPO. A larger discount factor in principle enables the algorithm to seek a policy that performs well for the average-reward criterion. But, unfortunately, a larger discount factor can also increase the variance of the gradient estimator (Zhao et al., 2011; Schulman et al., 2016) and degrade generalization (Amit et al., 2020). Moreover, algorithms with discounting become unstable as $\gamma \to 1$ (Naik et al., 2019). The discount factor therefore serves as a hyperparameter which can be tuned to improve performance. This is supported by the observation that the optimal discount factor is different for each environment (0.999, 0.99, 0.95 for HalfCheetah, Ant, and Humanoid respectively), where choosing a suboptimal discount factor can have significant consequences. (For Ant and Humanoid, the optimal discount factor is 33.9% and 65.6% better than the second best discount factor.) We have shown here that using the average reward criterion not only delivers superior performance but also obviates the need to tune the discount factor.

To further support our conclusion, we will also compare ATRPO and TRPO using an alternative evaluation protocol. In this protocol, after every one million samples of training we run 10 separate evaluation trajectories of fixed length 10,000 time steps using the current policy with no exploration. The random seeds used for evaluation are different from those used in training. Figure 2 shows the average reward of these trajectories, Once again ATRPO provides superior performance.

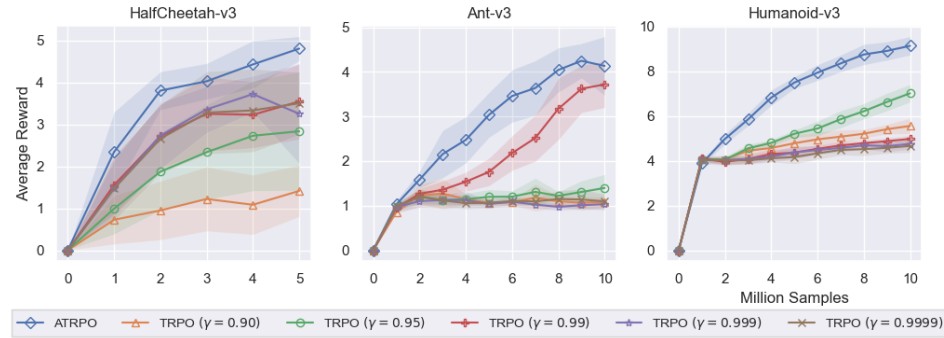

Figure 2: Comparing performance on evaluation trajectories of length 10,000. For each random seed used in training, we use a different unseen random seed to run 10 test trajectories after every 1 million samples of training. The solid line is averaged over these unseen random seeds. The shaded area is one standard deviation.

## 8 CONCLUSION

In this paper, we introduced a novel policy improvement bound for the average reward criterion. The bound is based on the average divergence between two policies and Kemeny's constant. We showed that previous existing policy improvement bounds for the discounted case results in a non-meaningful bound for the average reward objective. Our work provided the theoretical justification and the means to generalize the popular trust-region based algorithms to the average reward setting. We demonstrated through a series of experiments that our method is highly effective on high-dimensional continuing control tasks. In particular, we showed that when the natural objective of the task is undiscounted, discounting can lead to suboptimal behavior. To the best of our knowledge, we are one of the first to address how DRL methods can be used to learn undiscounted continuing control tasks with large state and action spaces.

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
