# OpenReview forum: "Average Reward Reinforcement Learning with Monotonic Policy Improvement"
_ICLR.cc/2021/Conference — Reject_

### Official Review · AnonReviewer1 · 2020-10-23
**Average Reward Reinforcement Learning with Monotonic Policy Improvement**

**Rating:** 6
**Confidence:** 4

**Review:**

***Summary***
The paper proposes an extension of the performance improvement bound, introduced for the first time by Kakade & Langford (2002), to the case of average reward performance index instead of discounted return. The paper starts with a theoretical contribution in which all the steps of the original derivation are adapted for the new setting, leading to a new performance improvement bound. Then, this theoretical achievement is employed to derive the corresponding adaptation of TRPO (called Average Reward TRPO) and CPO (called Average Cost CPO). Finally, an experimental evaluation on some Mujoco domains is provided.

***Major***
- (About the fundamental matrix) The main difference between the performance improvement bound for the discounted return and the one presented in this paper is the constant that multiplies the expected total-variation divergence between the policies. While for the discounted case, this constant is a function of the discount factor and of the maximum advantage, in the average reward case it depends on the maximum advantage as well and on the norm of the fundamental matrix of the Markov chain. The authors clarify that this matrix is guaranteed to exist for regular chains. However, the value of its norm, although finite, can be very large. From a practical perspective, this has limited implications as the objective that is actually optimized translates the penalization into a constraint, ignoring this constant. Nevertheless, it would be interesting if the authors could elaborate more on the properties of this matrix, maybe providing some bounds of its norm, if possible, in terms of more manageable quantities.
- (Organization) I have some concerns about Section 5.2. First, the change of perspective from rewards to costs might not favor the clarity of the paper. Second, concerning Average Cost CPO, no experiments are reported in the main paper but in the appendix only. Moving also Section 5.2 to the appendix and reserve more space for the proofs of the theoretical results in the main paper might improve the organization.
- (Experiments) I have some concerns about the interpretation of the experimental results, especially on how ATRPO is compared to TRPO. The two algorithms are designed to optimize different objective functions: the average reward and the discounted return, respectively. The plots that are shown in Figures 1 and 2 report the "Return", which I assume to be the undiscounted sum of the rewards over 1000 and 10000 steps respectively. I am wondering whether drawing conclusions based on this performance index is meaningful. Indeed, optimizing the undiscounted sum of the rewards is closer to optimizing the average reward rather than the discounted return and I am not surprised that ATRPO outperforms TRPO. Can the authors elaborate on this point and explain why they think that the comparison is fair?

***Minor***
- In Section 2, the paper requires that the state and action spaces are finite. Is this assumption really necessary to derive the presented results? In the experimental evaluation section, the algorithms are tested on the Mujoco suite that is made of continuous state-action environments.
- Equations (9), (10), (12) sometimes the state below the expectation is bold.

***Overall***
I think that the paper can be considered incremental compared to TRPO and, more in general, to the papers that study the performance improvement bounds in the discounted setting.  Moreover, I have some concerns about the interpretation of the experimental results. Therefore, my current evaluation is borderline.

---

> ### Author Response · Authors · 2020-11-19
> **Reply to AnonReviewer1**
>
> We are very grateful for your thoughtful comments and suggestions.
>
> We appreciate your comment on the fundamental matrix, we hope our new and improved bound and additional analysis on Kemeny’s constant has helped to alleviate some of your concerns.
>
> With regards to the experiments. Yes, you are correct that the y-axis is the undiscounted sum of rewards (we will make this clear in our revision). The trajectory is always summed over a fixed length of 1000 and 10,000, so the undiscounted sum of rewards and the average rewards are equivalent in this case. We wish to point out that we do not believe this metric penalizes the discounted case since even though traditional algorithms evaluating the MuJoCo benmark are trained using a discounted objective, they are always evaluated on the undiscounted sum of rewards (see e.g. Schulman et al. (2015), Wu et al. (2017), Schulman et al. (2017), Vuong et al. (2017)).
>
> You are also correct in pointing out that the state and action space in MuJoCo is continuous, and therefore violates the assumption in our theory. But to the best of our knowledge, there are no theoretical results regarding policy improvement on continuous spaces even though the community has demonstrated satisfactory empirical performances on these domains. We believe it would be interesting to see theoretical work in this area which may bring about additional insights.
>
> We would also like to thank the reviewer for pointing out the typo in some of our equations.
>
>
> References
>
> Schulman, J., Levine, S., Abbeel, P., Jordan, M., & Moritz, P. (2015). Trust region policy optimization. In International conference on machine learning (pp. 1889-1897).
>
> Wu, Y., Mansimov, E., Grosse, R. B., Liao, S., & Ba, J. (2017). Scalable trust-region method for deep reinforcement learning using kronecker-factored approximation. In Advances in neural information processing systems (pp. 5279-5288).
>
> Schulman, J., Wolski, F., Dhariwal, P., Radford, A., & Klimov, O. (2017). Proximal policy optimization algorithms. arXiv preprint arXiv:1707.06347.
>
> Vuong, Q., Zhang, Y., & Ross, K. W. (2019). Supervised policy update for deep reinforcement learning. Internal conference on learning representations.

---

> > ### Comment · AnonReviewer1 · 2020-11-21
> > **Reply to Authors**
> >
> > Thank you for your reply. I appreciate the clarification about the fundamental matrix and the introduction of the Kemeny’s constant. Concerning the experimental evaluation, I am still not convinced that the comparison is completely fair since the discounted objective is not the one that is used to compare the algorithms. I see, as the authors point out, that optimizing a discounted objective and evaluating on the undiscounted objective is a common (but inappropriate in my opinion) practice. Nevertheless, my opinion on the paper is positive.

---

> > > ### Author Response · Authors · 2020-11-22
> > > **Reply to Reviewer1**
> > >
> > > Thank you so much for your reply, we appreciate your comments. We understand your concerns regarding the experiments. You pointed out that it is inappropriate to optimize using the discounted objective and evaluate on the undiscounted objective. We completely agree and this is in fact one of the issues we wanted to address in our paper. We will also clarify this in our revision.
> > >
> > > When presented with an RL problem, we believe the choice of the objective function (discounted or average reward) should depend on which objective function more accurately describes the actual goal of the problem we are trying to solve.
> > >
> > > In certain applications, discounting is appropriate for the problem description. One example is when the reward corresponds to financial gains, in this case it makes sense to give more weight to money earned more recently. The discount factor here has a natural interpretation as it relates to the interest rate. However in MuJoCo, the undiscounted objective more accurately describes the goals of the problem (i.e. an agent’s performance w.r.t. the reward signal should be equally important at time step 1000 as it is at time step 1). The same can be said for the vast majority of benchmarks used for evaluating DRL algorithms. In the case of TRPO (and similarly many other DRL algorithms), discounting is used often for mathematical convenience. Prior to our work, there has been no theoretical or empirical evidence to support applying trust region methods to the average reward. In general, there's a lack of useful DRL methods that deals with the average reward objective. We hope to demonstrate in our paper that when the actual objective we want to evaluate is undiscounted, discounting is unnecessary and may lead to suboptimal performance and we are in fact better off using the average reward objective.

---

### Official Review · AnonReviewer4 · 2020-10-24
**undiscounted version of TRPO**

**Rating:** 5
**Confidence:** 4

**Review:**

This paper identifies an important problem: policy optimization in undiscounted continuing tasks. This is indeed important since the discounting factor may not be appropriate in certain applications such as health care or robotics. I feel the main contribution of this paper is Theorem 1, an average reward version of the monotone improvement theorem. Naively applying the discounted version results in a trivial bound.

My concern is about the algorithm (13) that is almost the same as TRPO, with a bit straightforward extension on the estimation for the advantage function. In the constrained form, the KL radius is viewed as a tuning parameter but in the regulaized version (12), the \xi involves a complicated form involving true P to guarantee the monotone improvement. This does not make very much senses to do this approximation. Note that in TRPO, there is no true P involved.

A large body of policy optimization works that focus on average-reward is perhaps missing. 1. Online Markov Decision Processes. (2009). 2. Online markov decision processes under bandit feedback (2010). Those two assumes the P is known but the average reward performance difference lemma has been explicitly derived in those two. More recently, "POLITEX: Regret Bounds for Policy Iteration Using Expert Prediction" considered a fully RL setting and average-reward setting with guarantee.

====================
Thanks the authors' response. Based on the originality of Theorem 1, I increase my score by 1 but still a bit worry about the sufficient algorithmic contribution beyond TRPO.

---

> ### Author Response · Authors · 2020-11-19
> **Reply to AnonReviewer4**
>
> We thank the reviewer for your thoughtful comments.
>
> In response to your comment on having $P$ in the bound. In the discounted case, not having the transition matrix appear in the bound is only possible since a discounted MDP is not truly infinite-horizon and can be shown to be equivalent to a related finite horizon problem (see Proposition 5.3.1 of Puterman (1994)). The discount factor can be used to control the effective horizon where larger discount factors correspond to longer horizons. For the average reward setting, in particular with regards to Lemma 3 which led to Theorem 1, the sensitivity of the state occupancy distribution w.r.t. the policy can vary depending on the chain structure and long-term behavior of the underlying Markov chain. Hence it is only natural that the multiplicative constant in Lemma 3 depends on the transition matrix.
>
> We also wish to point out that for the parallel bound for Theorem 1 in the discounted case, the multiplier in front to the TV divergence is not a simple constant. It involves an additional $\max_s E_{\pi’}|A^{\pi}(s,a)|$ term which involves both the current policy and the new policy. In both the discounted and the average reward cases, the trust region approximation is not ideal but does actually work quite well in practice. We therefore do not believe having the transition matrix presents any issues here.
>
> We would like to thank the reviewer for pointing out additional work related to our paper, we will be including the two papers on online MDPs in our discussion of the average reward performance difference lemma in our subsequent revision. However, POLITEX was already included and we have noted their contributions in the original submission of our paper (see Section 6).
>
> References
>
> Puterman, M. L. (1994). Markov decision processes: discrete stochastic dynamic programming. John Wiley & Sons.

---

### Official Review · AnonReviewer3 · 2020-10-27
**Shows strong signs of a good paper, but a few questions remain about the empirical evaluation.**

**Rating:** 6
**Confidence:** 3

**Review:**

This paper advocates for the use of the average reward objective in long horizon, continual reinforcement learning settings, and it examines whether a limiting argument for the discount is sufficient for extending theoretical results from the discounted setting to the average-reward setting. In the case of policy optimization, the paper shows the monotonic improvement bound from Schulman et al. (2015) becomes vacuous as gamma approaches one. The rest of the paper presents theory to support a new, non-vacuous bound, which leads to a new algorithm analogous to TRPO (A-TRPO). The paper additionally shows how their main theorem can be applied to constrained MDPs in Section  5.2. Empirical results are provided in three MuJoCo domains modified to provide non-episodic experience.

I found many strengths in the paper. For one, it was enjoyable to read and fairly well organized. The authors presented their ideas clearly, and they appeared logically-connected throughout the paper.

The paper is also well motivated. Continual reinforcement is a broad area of open research where new advancements stand to improve the generality of our current AI/ML systems. In that setting, the average reward objective for MDPs seems most natural. But despite the existence of average-reward RL algorithms, many of them are not well theoretically well understood (e.g. R Learning). And as the authors bring up, it is not known if more current theoretical results about policy optimization apply to the average-reward setting.

There are few minor issues I have with the presentation, and some questions I have regarding the experimental results. Pending a satisfactory response to these issues, which I describe below, this paper could be ready for publication.

Although the results of Section 5.2 seem useful for those interested in constrained decision making processes, they also seem fairly anecdotal in relation to the paper’s main claims and for what the main paper presents. The claims supported with substantial empirical evidence only use A-TRPO.  The A-CPO results have been tucked away in the appendix.  I suggest the authors remove this material, since the A-TRPO results are sufficient to support your main claims, and considering the constrained MDP setting in addition to average reward and discounted is too much for the length of a conference paper

Section 7 presents empirical data showing the performance of the proposed algorithm (A-TRPO) and compares it to TRPO with different discount factors. The presented results seem positive: suggesting that A-TRPO finds useful policies within the considered data regime. However, this section is terse, and so I have a few questions for the authors:

I’m not yet convinced the evaluation is entirely sound. The performance of an average reward agent is compared to discounted agents using their respective returns averaged over the last 1000 steps of their 100 most recent trajectories. I take issue with this approach for several reasons:

1. It is not clear that it has a meaningful connection to the average-reward criterion. Why not use the average reward, which is what the paper advocates for as the natural metric in continual settings?
2. Restricting the evaluation to the last 1000 steps, or trajectories of length 1000, (authors please specify) biases the outcome against discounted agents whose effective horizon is 1000 or greater. These agents are trained to maximize cumulative reward on horizons greater than or equal to 1000, but then they are evaluated on horizons less than that.
3. In a continual setting, there is no separate evaluation phase; learning continues indefinitely. Therefore, it seems more appropriate to report the reward averaged over a horizon where sufficient mixing has occurred, rather than report offline evaluation performance.

It is not immediately clear to me that the modifications to MuJoCo result in a well posed continual learning problem. Can the authors please clarify if the maximum episode length cap was removed for training?  Can they also clarify exactly how the start states were sampled? Are there any aspects that change or remain the same between falling down and restarting? A complete characterization of the factors of variation would be helpful here.

I do not have a good sense for what data is being plotted in Figures 1 and 2. The data seems to vary between the plot markers. So my question is: what do the plot markers represent? Has this data been smoothed at all? If so, then can you please provide more details so we can understand how the data was processed.

The paper does a good job of positioning its contribution with respect to prior work throughout the text. Though it wouldn’t hurt to expand the Related Work section so it reads less like a list of facts. The goal would be to help readers understand the contribution by comparing and contrasting it with what others have done.

It would be helpful to include a video showing the execution of the policy as it is being learned. This would allow reviewers to verify if the learned average-reward polices are actually useful, and they do not simply achieve better evaluation metrics than the presented baselines.

The questions identified in the beginning of Section 7 could be more specific. For example: “Which performance criterion is more suitable for continuing control tasks?”. This question strikes me as something that has been answered in the affirmative throughout the long history of average-reward and infinite horizon discounted MDPs. This is also mentioned in two references you provide (Sutton & Barto, and Naik et al. 2019). Could you be more specific about what you are trying to address here, how this is different than what others have shown, and about how Question 2 is different from Question 3.

Many of the theoretical results cite a set of class notes (Achiam 2017). I suggest the authors remove this citation and reproduce the results needed for their arguments in the appendix. This provides a way for those to be peer reviewed and for future work to reliably cite them.

---

> ### Author Response · Authors · 2020-11-19
> **Reply to AnonReviewer3**
>
> We want to thank the reviewer for your detailed comments. We have incorporated your suggestions into our paper, in particular into our empirical evaluations. As you requested, we will provide additional details on the modified environment.
>
> In our empirical evaluations,  we had two goals: 1) we wanted to demonstrate through a fair comparison that using discounting could potentially be problematic in scenarios with extremely long time horizons and 2) we wanted to put discounted TRPO in the best light possible and compare it with ATRPO, so that we can better understand how the performance for the average reward differs when controlling all other factors. Many of our design choices were motivated by these two goals. We will make this clear in our revision, and be more specific in Section 7 on the questions we attempt to answer. Below we address some of the specific concerns you raised regarding the implementation (which we will also include in our revision):
> - Since we used fixed trajectory lengths of 1000 during training, the total sum of rewards we used for the y-axis is equivalent to the average reward.
> - During training, at iteration k, we collect a batch of data consisting of 5 trajectories each of length 1000, making up a total of 5000 data points per batch. We did not want to unnecessarily penalize the discounted algorithm since we noticed that collecting multiple trajectories per batch boosted the performance of discounted TRPO. The choice of batch size and episode length were all made to put the discounted algorithm in the best possible light. Also using the total reward averaged over the last 100 trajectories to evaluate the algorithm is standard practice in many on-policy algorithms (see e.g. Schulman et al. (2017), Wu et al. (2017), Vuong et al. (2019)).
> - We included a separate evaluation phase because we wanted to understand the behavior of policies trained under the discounted and undiscounted algorithm over long horizons, or as you pointed out, after sufficient mixing has occurred, since during our training phase we only ran trajectories for a maximum length of 1000.
> - However we do recognize your concerns and we understand that for continuing tasks it would make more sense to not use separate trajectories. In our revision, we will include additional experiments where each batch of data only consists of one continuous trajectory (rather than five different trajectories), you will notice that in the new experiments the performance gap between ATRPO and TRPO is much greater.
>
> The plot markers were placed at fixed intervals to label the different curves so that our plots are colorblind-friendly (i.e. they can be distinguished by both the colors and markers); they do not have any meaningful connection to the data in the plot. The plots were smoothed in the sense that they are averaged over the last 100 trajectories encountered by the agent; however, as we have noted earlier, this is standard practice for on-policy algorithms.
>
> We appreciate your suggestion on our related work section. We will rewrite parts of this section to better compare with the contributions of our work.
>
> There is some confusion regarding the citations which you pointed out. In our references, we included two works by Joshua Achiam. One is the paper Constrained Policy Optimization (cited as Achiam et al. 2017), which is an important paper on the policy improvement theorem in the discounted case. We cited this paper multiple times in our theoretical results in making parallels between the average reward and discounted case. In the appendix, we cited a set of lecture notes from Joshua Achiam (cited as Achiam 2017) as a source for detailed pseudocode for the discounted TRPO algorithm. We did not cite these lecture notes in our theoretical results. We apologize for the confusion, in our revision we will make clear the distinction between the two works.
>
> References
>
> Wu, Y., Mansimov, E., Grosse, R. B., Liao, S., & Ba, J. (2017). Scalable trust-region method for deep reinforcement learning using kronecker-factored approximation. In Advances in neural information processing systems (pp. 5279-5288).
>
> Schulman, J., Wolski, F., Dhariwal, P., Radford, A., & Klimov, O. (2017). Proximal policy optimization algorithms. arXiv preprint arXiv:1707.06347.
>
> Vuong, Q., Zhang, Y., & Ross, K. W. (2019). Supervised policy update for deep reinforcement learning. Internal conference on learning representations.

---

### Official Review · AnonReviewer2 · 2020-10-28
**Interesting paper with practical significance for average reward policy gradient setting**

**Rating:** 6
**Confidence:** 5

**Review:**

This paper proposes a practically feasible algorithm for the average reward setting in RL. The key idea is to propose a lower bound policy improvement objective, for the average reward setting compared to the more popular and general discounted reward setting. This work follows directly from previous impactful works on policy gradients with trust region optimization (TRPO) and shows why previous approaches cannot be directly applied for the average reward criterion. This is a significant contribution in RL research, for the average reward performance criterion, and opens doors for future works in this direction that might be impactful to the community. Overall this is a good paper. There are few major and minor concerns I have about the contribution, mostly from an algorithmic point of view, which are described in details below.

Overall comments :
- This work provides monotonic performance improvement guarantees, by deriving a lower bound policy improvement criterion for the average reward setting. It is well known, even though less significantly realized that the existing policy improvement theorem for the discounted reward setting leads to a trivial bound for the average performance criterion. This work builds from that and derives a novel result which can bound the average reward performance objectives based on the divergence between two policies, as mostly done in a lot of policy optimization algorithms (e.g TRPO and many variants building from TRPO).
- Additionally, the authors propose a constrained optimization variant of their algorithm, building from the CPO algorithm (Achiam et al) and shows that the average reward criterion can also be used for policy optimization under cost constraints.
- The paper is well written and easy to follow, with the key theoretical contributions clearly defined. It builds from the well known proposition from Achiam et al., for the upper and lower bounds for the performance difference lemma (Proposition 1). Lemmas 1-3 are known results, adapted for the average reward setting, including the performance difference lemma for the average reward objective.
- The key result is shown in theorem 1, which bounds the performance difference based on the average reward criterion, but reduces to the divergence between policies only. Equations 9 and 10 are extensions from Achiam et al., adapted for this setting, with the typical assumption that the changes in occupancy measures between $\pi'$ and $\pi$ are insignificant. This is a standard assumption, assuming minimal state distribution shift between the two policies, as typically done in TRPO, CPO and variants.
- Experimentally this work nicely demonstrates the significance of the average reward criterion in continuous control tasks, which are adapted for a continuing setting, instead of the typical episodic setting with reset states. Under this modification, the authors compare to TRPO and CPO with different discount factors in the continuing environment, and shows the significance of optimizing the average reward lower bound objective. The authors do not compare to other well known policy gradient baselines, which I believe is fine in this case; since the key contribution is to propose an algorithm applicable for continuing environments - and modifications on top of their ATRPO algorithm can be made, as required, to compare to other well known existing baselines. Even though the experimental results are less exhaustive, I think this is not too of a problem, compared to the significant algorithmic contribution this paper introduces.

Few issues and comments for improvement :
I think the major drawback or clarity this paper requires is to describe their algorithmic steps more clearly. At first glance, there are few issues which seems unavoidable, and it would be better if the authors can clarify on these. These are as follows :
- Lemma 3 bounds the divergence between stationary distributions in terms of the divergence between policies. This result is adapted, as shown in appendix, following from Achiam et al., and is known from other papers that such divergence between stationary distributions can be bounded. Lemma 3, equation 8 shows that this bound depends on the l-infinity norm between the transitions under the policies. Proof of Lemma 3 is easy to follow. My major concern is how is this term in equation 8 dealt with in the overall algorithm?
- Theorem 1 following from the stated lemmas, depends on this, as highlighted in the $\epsilon$ term and in Algorithm 1 (equation 12). It seems that the assumption that the stationary distributions between \pi' and \pi are close in this case is less justified in the average reward setting compared to the discounted setting. Can the authors comment on this? In the discounted setting, the normalized occupancy measures between two policies can perhaps be assumed to remain close, as done in TRPO. However, does the same hold for stationary distributions? If so, why would that be? I think this is a major assumption.
- I do see why in the overall algorithm, this assumption is made; since otherwise finding the sensitivity of stationary distributions w.r.t changes in policy parameters might be difficult to compute; but theoretically, it would be better if authors can comment on it.
- Section 5.1 outlines the key algorithm and steps. Equation 13 shows the overall objective with KL constraints, which at first glance, is almost equivalent to the discounted reward setting. However, as pointed out later, the advantage function takes account of the average reward, and the overall algorithm can be naturally extended from TRPO and CPO. However, I think it would be better if the authors comment on the above issues, and how is the overall algorithm implemented. In appendix C, there are discussions on the critic estimate, if this was to be extended in an actor-critic setup with lambda returns - however, appendix C does not really give anything meaningful as expected. It seems that the overall difference is in equation 33, where the target is now modified with the average reward. This makes the algorithmic contribution clean, but I wonder how is this implemented in practice. The target now requires computation of the average reward, for every s,a pairs? This seems to be a bottleneck, and comes to the major drawback for the average performance criterion anyway? How is this avoidable?
- Overall, I think the algorithmic implementation of this is not clearly explained. It seems there are major steps ommited in the overall description of the algorithm - and this is partly also because the authors propose ATRPO and ACPO algorithms, which are two different contributions itself. I think it would be better if the key algorithmic idea and implementation details are rather included in the main text, so that the significance of the work can be better highlighted; and perhaps the extensions with cost constraints can be moved to the appendix.

Summary : I think overall this is a good paper, with some clarities that are still required from the authors. This is a good contribution, extending existing impactful works from the discounted reward setting to the average reward setting. The authors propose theoretical justifications as well as a practically feasible algorithm. If the authors can clarify some of the major concerns I have, given my understanding is correct, I am willing to further increase the score. However, as it is, I would recommend marginal acceptance, and open to discussions with other reviewers and authors to clearly understand the significance of the work.

---

> ### Author Response · Authors · 2020-11-19
> **Reply to AnonReviewer2**
>
> Thank you for your very detailed comments. Your suggestions have been extremely helpful in improving our manuscript. Here we will address some of your comments which we have not yet addressed above.
>
> You are correct in pointing out that in certain cases the stationary distributions could be quite sensitive to changes in the policy, we hope our updated bound for Lemma 3 based on Kemeny’s constant has provided more insight on when the stationary distributions can be assumed to be close.
>
> However we also wish to note that in the discounted case, the assumption that the normalized state occupancy distributions remain close can also be violated.  Achiam et al. (2017) showed a similar result to Lemma 3 in the discounted case where the change in future discounted state visitation distribution is bounded in terms of the change in policy up to a multiplicative constant which depends on the discount factor ($2\gamma / (1-\gamma)$ to be precise). It can be easily shown that this multiplicative factor goes to infinity as gamma goes to 1 meaning that it is not useful for long horizon problems.
>
> In our revision, we will also be providing pseudocode for our algorithm and additional implementation details, especially on the implementation of the critic. You pointed to a potential bottleneck in our algorithm in that the target used by the critic requires calculating the average reward. This is in fact not the case since our algorithm is on-policy and we only need the average reward for the current policy. In our algorithm, we collect samples from the environment using our current policy which we store in a buffer; it takes little extra effort to calculate an average of the reward for the policy using samples we have stored, which we then use to update our critic and policy. After we update our policy and critic using these samples, we discard all old samples and collect new data using the updated policy and repeat. We hope that the pseudocode we will provide in our revision will provide additional clarity.
>
> References
>
> Achiam, J., Held, D., Tamar, A., & Abbeel, P. (2017). Constrained policy optimization. International conference on machine learning.

---

### Public Comment · ~Bogdan_Mazoure1 · 2020-11-16
**Potential similar results to Lemma 3 in related work**

This is an interesting paper for the average reward policy proposing an improvement step using a lower-bound. The average reward scenario indeed allows for a nicer analysis in terms of Markov chain long-run distributions for the case of a fixed policy. I believe that Lemma 3 and Thm. 1 are present in a similar form in this paper [1] [Representation of Reinforcement Learning Policies in Reproducing Kernel Hilbert Spaces](https://arxiv.org/abs/2002.02863v1) , which I think is worth mentioning, as it would provide a more thorough overview of the related works to the reader for the case when the policy is fixed.

---

> ### Author Response · Authors · 2020-11-19
> **Thank you**
>
> Thank you so much for pointing us to this very interesting paper, we will add a comparison to this work in our related work section.

---

### Author Response · Authors · 2020-11-19
**Response to all reviewers**

We would like to thank all the reviewers for your detailed and thoughtful feedback. We are grateful that all reviewers recognize our contribution in extending the popular trust-region based methods to the average reward criterion, with several reviewers pointing out the importance of continuing control tasks in RL research. Much of the recent progress made by DRL methods has been on the discounted criterion; it is our hope that our work will help promote more interest in the average reward criterion and continuing control in DRL. Several reviewers have found our work to have solid theoretical grounding and strong empirical results. We have incorporated many of the suggestions made by the reviewers into our revised work and are in the process of finalizing that revision which we will upload shortly.  Here we will address some common concerns raised by multiple reviewers. We will also be responding to each reviewer separately to address their remaining concerns and comments.

Lemma 3 in our paper dealt with how sensitive the change in stationary distribution is w.r.t. to changes in the policy. Several reviewers (Reviewer2, Reviewer1) raised concerns on the interpretability of Lemma 3 (and by extension Theorem 1), specifically regarding the multiplicative factor involving the infinity norm of the fundamental matrix. Both Reviewer2 and Reviewer1 pointed out that the multiplicative factor could be very large.

To help alleviate the interpretability issue, we now use the relationship between the fundamental matrix and mean passage time to derive a more interpretable bound for Lemma 3, where we replaced the infinity norm with a constant factor known as Kemeny’s constant (Grinstead and Snell, 2016). This constant is equal to the trace of the fundamental matrix and has an intuitive interpretation: it is the expected number of steps it takes for an agent starting at a start state to reach some goal state, where the goal state is drawn from the stationary distribution. For very ‘well-connected’ Markov chains where an agent can easily and quickly get to any state, this constant is small and stationary distributions are therefore not very sensitive to changes in policy. However for Markov chains that are ‘nearly multichain’  (but nevertheless still unichain), the factor can become very large. We will be adding more analysis regarding Kemeny’s constant in our revision.

Several reviewers asked for more implementation details, specifically regarding the critic estimator (Reviewer2) and environment (Reviewer3). In our revision, we will be adding additional details on the critic estimator as well as pseudocode on how it is actually implemented in practice, which we hope will clear up any confusion the reviewers may have. We would also like to point out that even though the only difference between ATRPO and TRPO is the critic estimator, the critic estimator in the average reward case is not simply setting the discount factor to 1 and is more involved. We will also be adding additional details on the modified MuJoCo environment as requested by Reviewer3.

Reviewer2, Reviewer3, and Reviewer1 all recommended moving the constrained RL section to the appendix. We will be making such an arrangement in our revision as well as adding additional material to the main text as was recommended by the reviewers. We will also be expanding a little in our revision on our reasoning for including the constrained RL material. In non-constrained RL, even if our goal is to maximize an undiscounted objective, in some cases maximizing a discounted objective can be used as a proxy since increasing the value for one could also increase the value for the other. However in the constrained setting, most algorithms require us to evaluate the actual value of the constraints and in these instances it is often never the case that we can use a discounted constraint when the actual constraint is undiscounted.


References

Grinstead, C. M., & Snell, J. L. (2012). Introduction to probability. American Mathematical Soc..

---

### Author Response · Authors · 2020-11-25
**Revision Summary**

We have just uploaded a revised version of our paper. We would like to once again thank all the reviewers for the thoughtful suggestions you have made. Your suggestions have been invaluable in helping us to improve our manuscript. Below is a summary of the changes we have made:

Based on suggestions from Reviewer1 and Reviewer2, we modified the bound in Lemma 3 and Theorem 1 using relationship between mean first passage time and the fundamental matrix. The multiplicative factor now depends on Kemeny's constant. We added a new proof for Lemma 3 in the appendix, as well as additional discussion on Kemeny's constant – i.e. when is the factor large or small; for what kind of MDPs do our assumptions hold well – in the main text

As recommended by Reviewer2, we added a new section (Section 5.2) in the main text going over implementation details with pseudocode for the ATRPO algorithm.

For our experiment section, we modified the experiment so that for each batch of data, instead of running 5 separate trajectories, we now use one continuous trajectory. We also rewrote this section to better motivate our empirical evaluations. In the appendix, we added additional details on the modified environments. These changes were based on suggestions and comments made by Reviewer1 and Reviewer3.

As Reviewer3 suggested, we also rewrote parts of our related works section to better compare with our contributions.

We also moved the Constrained RL section to the appendix based on the recommendation of Reviewer1, Reviewer2, and Reviewer3.

We added several additional references related to new bound in Lemma 3 and two additional papers kindly pointed out to us by Reviewer4 on the policy difference lemma for the average reward.

We also fixed some typos from our original submission.

Overall we believe our paper is very important to the community, both in terms of our contributions and in bringing attention to DRL methods for continuing control and the average reward criterion. If you are happy with our response and revision, we would be grateful if you would consider recommending acceptance for our paper.

---

### Decision · Program_Chairs · 2021-01-07
**Final Decision**

**Decision:**

Reject

**Comment:**

This paper proposes an extension of the monotonic policy improvement approach to the average reward case.
Although the reviewers acknowledge that this work has merits (well written, clearly organized, well-motivated, technically sound) the reviewers have raised several concerns, which have been only partially addressed by the authors' responses. In particular, Reviewer4 is still concerned about the discrepancy between the theorem and the implemented algorithm, and the proposed simplification used in the implementation boils down to an algorithm that is very similar to TRPO, thus making the contribution quite incremental as also stressed by Reviewer1. Furthermore, I share the concerns raised about the fairness of comparing algorithms that optimize different objective functions.
I suggest the authors take into serious consideration the suggestions provided by the reviewers in order to produce an improved version of their work.
The paper is borderline and I think that it needs another round of fresh reviews before being ready for publication.